# Expression of Sigma-Class Glutathione-S-Transferase in Fetal and Pediatric Filum Terminale Samples: A Comparative Study

**DOI:** 10.3390/medicina55050133

**Published:** 2019-05-13

**Authors:** Cahit Kural, Serpil Oguztuzun, Gülçin Güler Şimşek, Servet Guresci, Pınar Kaygın, Soner Yasar, Ozkan Tehli, Yusuf Izci

**Affiliations:** 1Department of Neurosurgery, University of Health Sciences, Gulhane Education and Research Hospital, 06010 Ankara, Turkey; cahitkural23@gmail.com (C.K.); dr.soneryasar@gmail.com (S.Y.); ozkantehli@gmail.com (O.T.); 2Department of Biology, Kırıkkale University, 71450 Kırıkkale, Turkey; soguztuzun@yahoo.com (S.O.); pinar.kaygn@gmail.com (P.K.); 3Department of Pathology, University of Health Sciences, Keçiören Education and Research Hospital, 06000 Ankara, Turkey; drgulcinguler@gmail.com; 4Department of Pathology, University of Health Sciences, Numune Education and Research Hospital, 06230 Ankara, Turkey; gurescisa@hotmail.com

**Keywords:** tethered cord syndrome, glutathione-S-transferase, child, surgery

## Abstract

*Background and objectives*: The pathophysiology of tethered cord syndrome (TCS) in children is not well elucidated. An inelastic filum terminale (FT) is the main factor underlying the stretching of the spinal cord in TCS. Our study aimed to investigate the expression of glutathione-S-transferase (GST) in children and fetal FT samples in order to understand the relationship between this enzyme expression and the development of TCS. *Materials and Methods:* FT samples were obtained from ten children with TCS (Group 1) and histological and immunohistochemical examinations were performed. For comparison, FT samples from fifteen normal human fetuses (Group 2) were also analyzed using the same techniques. Statistical comparison was made using a Chi-square test. *Results:* Positive GST-sigma expression was detected in eight (80%) of 10 samples in Group 1. The positive GST-sigma expression was less frequent in nine (60%) of 15 samples from Group 2. No statistically significant difference was detected between the two groups (*p* = 0.197). *Conclusions:* Decreased FT elasticity in TCS may be associated with increased GST expression in FT. More prospective studies are needed to clarify the mechanism of the GST–TCS relationship in children.

## 1. Introduction

Tethered cord syndrome (TCS) is a functional disorder of the spinal cord which is secondary to stretching of the spinal cord. Impaired oxidative metabolism is the pathophysiological basis of TCS [1,2,3,4,5]. The main structure anchoring the caudal end of the spinal cord, the filum terminale (FT), is abnormally short and/or inelastic [6,7,8]. Progressive neurological and urological dysfunctions and orthopedic deformities secondary to fixation or tethering of the distal spinal cord are the main characteristics of TCS. This syndrome is often seen in children, and both the diagnosis and treatment are controversial and can be difficult for neurosurgeons. Cutting of the FT and release of the spinal cord is the main strategy of surgical treatment for TCS [9,10,11]. This procedure can be performed either by open surgery or an endoscopic approach [9,11]. However, section of the FT may not provide any improvement in the urological outcome of patients with occult TCS [10].

Fixation, stabilization, and buffering of the distal spinal cord from traction are the functions of the FT [12]. This is a viscoelastic band. It allows movement of the conus medullaris during flexion and extension of the spine [7,8,13,14]. In the case of an inelastic FT due to fibrosis, fatty infiltration or abnormal thickening, caudal tension and traction may cause stress on the conus medullaris [4,9,15,16]. Impaired oxidative metabolism may develop secondary to this stress injury and is correlated with neurological or urological dysfunctions in TCS [4,5,10].

Glutathione-S-transferase (GST) enzyme activity is present in the human body. GSTs are divided into different families: cytoplasmic, mitochondrial, or membrane-associated proteins involved in eicosanoid and glutathione metabolism [17,18,19]. In particular, studies on the role of GST in human systems during stress deserve scientific attention. GST is a universal biomarker in many organisms. In addition, biosynthesis of GST can be triggered by different stress factors, such as fibrosis and traction [17,20,21].

Sigma-class GSTs belong to the cytosolic GST (cGST) family. These enzymes are widely distributed throughout nature [22]. They not only exist in humans, but also in mice, rats, chickens, insects, flatworms, and mollusks [17,22,23]. The sigma-class GSTs have an N-terminal thioredoxin-like domain which binds glutathione in an extended configuration at the active site [17,24]. It has been previously shown that sigma-class GSTs exhibit prostaglandin synthase activity [19]. This enzyme initiates the pathway that leads to the formation of the J2-series of prostaglandins. These are important signaling molecules because they are ligands for peroxisome proliferator-activated receptor gamma (PPARɣ) [19,25]. Prostaglandin D2 (PGD2) is an early-phase mediator in inflammation. It also functions like a trophic factor in the central nervous system. Moreover, PGD2 also has a role in smooth muscle contraction/relaxation and inhibition of platelet aggregation [26]. PGD2 inhibits transforming growth factor beta (TGF-β)-induced collagen secretion [27]. Overproduction of PGD2 secondary to high expression of sigma-class GST may contribute to fibroblastic activity in the FT which results in fibrosis of the FT and development of TCS.

In our study, we aimed to clarify the relationship between histologically abnormal FT and tethering of the spinal cord. Therefore, we investigated the expression of sigma-class GST in FT samples which were obtained from children who underwent surgery for TCS. In addition, we analyzed the expression of this enzyme in fetal FT samples and compared the results.

## 2. Materials and Methods

Ethical approval for this study was obtained from the National Ethics Committee (Approval No: 7/18.1.2010). Ten pediatric patients with a diagnosis of TCS underwent surgical treatment and cutting of the filum terminale and release of the spinal cord was performed in these patients (Group 1) (Table 1).

An FT sample was obtained from each patient for histological analysis (Figure 1A,B). We obtained informed consent from all patients’ families for the surgical treatment of the patient, as well as for the analysis of FT samples. In addition, fifteen normal human fetuses, who were premature stillborns, were dissected for the histological analysis of FT (Group 2).

In terms of the gender of the fetuses, eleven were female and four were male. The mean gestation age was 25.4 weeks (range of between 14 and 35 weeks). There was no spinal malformation in the fetuses. No sign or findings of spina bifida were observed on the backs of the fetuses. A wide lumbosacral laminectomy was performed in the midline in order to reach the dural sac. Using microsurgical techniques, the conus medullaris and FT were seen after the dural opening. About 1 cm long segments of FT were determined and obtained for histological examination (Figure 1C). General histological examination of FT was performed using haematoxylin-eosin (H&E) staining. In addition, these samples underwent immunohistochemical examinations. 

### Immunohistochemical Staining

FT samples were fixed in 10% buffered formalin. Then, they were embedded in paraffin blocks. Sections of 4 μm thickness were obtained. For the observation of general tissue morphology, one section was stained with H&E. Endogenous peroxidase activity was blocked by incubating the sections in 1% hydrogen peroxide (*v/v*) in methanol for 10 min at room temperature (RT) for immunohistochemical examination. Then, the sections were washed in distilled water for 5 min. In a domestic pressure cooker, antigen retrieval was performed for 3 min using 0.01 M citrate buffer (pH 6.0) (Boster Biological Technology, Pleasanton, CA, USA, Cat no: AR0024). The sections were transferred into 0.05 M Tris-HCl (pH 7.6) containing 0.15 M sodium chloride (TBS) following washing in distilled water. These sections were incubated for 10 min at RT with super block (SHP125) (ScyTek Laboratories, West Logan, UT, USA). Then, they were covered with the primary antibody diluted 1:50 for GST Sigma (GSTS) (Boster Biological Technology, Pleasanton, CA, USA, Cat no: A05745) in TBS at 4 °C overnight. The sections were incubated at RT for biotinylated link antibody (SHP125) (ScyTek Laboratories, West Logan, UT, USA) following washing them in TBS for 15 min. Finally, treatment was followed with Streptavidin/HRP complex (SHP125) (ScyTek Laboratories, West Logan, UT, USA). In order to visualize peroxidase activity in the tissues, diaminobenzidine was utilized. Nuclei were lightly counterstained with haematoxyline. The sections were dehydrated and mounted after this procedure. Positive and negative controls were included in each run.

Light microscopy of immunohistochemically stained sections was performed by two pathologists who were unaware of the patients’ clinical information. Distribution, localization and characteristics of immunostaining were recorded. A brown color in the cytoplasm and/or nucleus of cells was evaluated as positive staining. Scoring was also performed by observers unaware of the patient data. Scoring differences between observers were resolved by consensus. For GST-sigma antibody, the intensity of the reaction was determined as negative (−), weak (+), moderate (++) or strong (+++). The results of sigma-class GST expression were compared with a Chi-square test for both groups. A *p* value of less than 0.05 was accepted as statistically significant.

## 3. Results

Ten FT samples from children with TCS (Group 1) and 15 FT samples from normal human fetuses (Group 2) were evaluated with histopathological and immunochemical examinations. Fibrosis was observed in 8 samples in Group 2 (Table 2), while it was observed in all ten of the FT samples from Group 1(Table 1). Moreover, fibrosis was moderate and strong in three samples from Group 1 (Figure 2 and Figure 3).

However, it was mostly negative or weak in Group 2. In addition, connective tissue and ganglion cells were obvious in Group 2 (Figure 4). Weak expression sigma-class GST was observed in 6 (40%) of 15 Group 2 samples (Table 2). This expression was also mostly weak in Group 1, but frequent, in 8 (80%) of 10 samples. Although sigma-class GST expression was higher in Group 1, the difference was not statistically significant (*p* = 0.197) (Figure 5).

## 4. Discussion

Fibrosis and an inelastic structure of the FT were obvious in histological examination of ten children with TCS. In addition, sigma-class GST expression was frequent in FT samples from the children with TCS. Frequent expression of GST may contribute to the inelastic properties of FT in TCS. However, this difference was not statistically significant and this may be due to the low number of samples in each group.

The filum terminale is an important fibrous anatomical structure in the pathophysiology and treatment of TCS. It is crucial to understand the morphometric parameters, histological structure, variants, and imaging of the FT for better diagnosis or treatment of TCS [6]. The FT consists of the filum terminale internum (FTI) and the filum terminale externum (FTE) [8]. The FTI shows a bigger strain under weight loading compared with the conus medullaris. It protects the conus medullaris from traction, together with the dentate ligaments [13]. However, the FTE has no obvious effect on the conus medullaris or cauda equine [28]. The FT has an elastic structure, while elastic fibers are decreased in children with TCS [12,16]. The FT also contains peripheral nerve fibers which are cut during the surgery for TCS. However, this does not cause any neurological deterioration because these fibers are probably non-functional [15]. Cutting the FTI, removal of arachnoid bands and attachments, and release of the spinal cord is the main treatment strategy for TCS. This may be performed through the same surgical incision if the TCS is secondary to open spinal dysraphism, or via an endoscopic approach which provides a smaller skin incision, narrow durotomy, and minimal tissue damage [9,11]. Isolated transection of the FTE via an extradural approach for a patient with TCS has no significant effect on the release of the spinal cord [28]. In our patients with TCS, we performed an open surgical approach and released the spinal cord using microsurgical techniques and under intraoperative neuromonitoring. We did not perform isolated transection of the FTE in any patient and we did not use an endoscopic approach for TCS surgery.

Glutathione S-transferases are multifunctional enzymes that can be found in prokaryotic and eukaryotic cells [19]. Based on sequence identity, there are seven classes of mammalian cytosolic GSTs including alpha, mu, pi, sigma, theta, omega, and zeta [17,18,19]. Sigma-class GST is one of the largest GST subfamilies, and has multiple functions. It can be found in vertebrate and invertebrate animals. GST-sigma has been characterized as glutathione-dependent hematopoietic prostaglandin synthases which are responsible for the production of prostaglandins in mammals and parasitic worms [29]. These enzymes can inactivate lipoperoxidation products and catalyze some biosynthetic reactions. They also play important roles in terms of structure and metabolism [27]. High levels of sigma-class GST expression in mammals, was found in the spleen, bone marrow, lung, placenta, adipose tissue, oviduct, and skin [19,22,23,25]. There is no study on the expression of sigma-class GST in the human spinal cord or filum terminale. Therefore, our study is the first to consider this enzyme expression in the FT.

GSTs are also important in the cellular uptake and intracellular transport of bilirubin [30]. Some isoforms of GST are specifically concentrated in neural cells [31]. Different types of stresses (e.g., variation in temperature, oxidative damage, and exposure to toxins) can regulate the expression of GSTs. In addition, antioxidant response elements can be found in cGSTs promoters [32]. GSH is a reactive oxygen species scavenger. GSTs are partly responsible for the metabolizing of oxidative stress products [17]. Moreover, there is increasing evidence that suggests omega, sigma, and theta class-GSTs have important roles in oxidative stress [23]. Sigma class-GST produces PGD2 [33]. PGD2 is actively manufactured in different organs such as the brain, spleen, thymus, bone marrow, uterus, ovary, oviduct, testis, prostate and epididymis. It has a role in many physiological processes. It has been shown that the hematopoietic PGD synthase belongs to the sigma-class GST family [25]. This enzyme is mostly observed in antigen-presenting, dendritic, Langerhans, Kupffer, megakaryoblastic, and mast cells. They probably have some functions in the production of PGD2 as allergic and inflammatory mediators [34]. As we know, an important histological finding of TCS is the progressive development of dense fibrous connective tissue or fibrosis in the FT. This causes loss of elasticity in the FT [1,7,12]. This might be secondary to an inflammatory response in the FT which may cause increased PGD2. Therefore, we investigated GST expression in the FT of the children with TCS and also in fetal FT samples. Tasaki et al. [26] showed that PGD2 metabolite delta 12-PGJ2 (9-Deoxy-delta 9, delta 12-13,14-dihydroprostaglandin 2D) enhanced collagen type I synthesis in human osteoblasts. However, there is no study on the effect of PGD2 on type III collagen. Type I collagen is the main collagen type that is synthesized by activated fibroblasts during the repair processes and the progression of fibrosis [30]. Fontes et al. [1] showed that the FT was composed of collagen type I and III collagen bundles on a matrix of elastin and elaunin fibers in adults. The elasticity of FT is probably provided by this structure. They also suggested that the changes in the type I/type III ratio or in the amount of elastic fibers may have a role in the development of a tethered cord [1]. Kural et al. [14] showed that collagen type III was mostly present in fetal FT, whereas type I collagen and elastic fibers were not detected. Therefore, it can be argued that the changes in type III collagen metabolism may play a role in the development of TCS by loosening the elasticity of FT. It was also previously shown that deterioration of collagen fibers in the FT, irregularity, deterioration of vascular structures, increase in adipose tissue and hyalinization can be seen in patients with TCS [1,5,7,12]. Therefore, we investigated sigma-class GST in FT samples to reveal the relationship between the elasticity of FT and the occurrence of TCS. The comparison of fetal and pediatric FT samples may help to clarify the pathophysiology of TCS.

Connective tissues are mainly formed by fibroblasts which can be found throughout the body. These cells are the principal source of the extensive extracellular matrix (ECM). Fibrosis is the thickening of the ECM secondary to an inflammation or physical injury. It plays a significant role in organ failure [20]. Idiopathic pulmonary fibrosis (IPF) is a lung disease characterized by excessive collagen production and fibrogenesis. In IPF patients, changes in glutathione redox status have been previously reported. Recently, McMillan et al. [35] showed that inhibition of pi-class GST may be used for the treatment of IPF. Similarly, fibrosis in the FT contributes to the development of TCS in children [1,30]. However, the development of fibrosis in the FT and its relationship with lipid peroxidation and collagen synthesis has not yet been elucidated. In our comparative study, we showed that sigma-class GST expression was observed in 9 (60%) of 15 fetal FT samples, while this expression was observed in 8 (80%) of 10 FT samples from children with TCS. It was clear that sigma-class GST expression was more prominent in the FT samples of patients with TCS. Therefore, it can be argued that high expression of GST may be associated with fibrosis of FT resulting in the development of TCS in childhood, and inhibition of GST may contribute to the improvement of TCS symptoms by decreasing fibrosis in the FT.

There are three limitations of this study. First of all, the number of samples in both groups is too low for more accurate conclusions to be drawn. Second is the lack of investigation of PGD2 to unveil the link between the development of fibrosis and the expression of GST. Third is the wide variation in the age of the patients with TCS; this variation may confound the results of GST expression, fibrosis, and structure of the FT. Studies with more FT samples and analysis of PGD2 may be helpful to better understand the correlation between fibrosis and TCS.

## 5. Conclusions

Although statistically insignificant, expression of sigma-class GST is high in the filum terminale of children with TCS. This may be associated with decreased elasticity of FT. Further histological studies are needed to elucidate the pathophysiology of TCS.

## Figures and Tables

**Figure 1 medicina-55-00133-f001:**
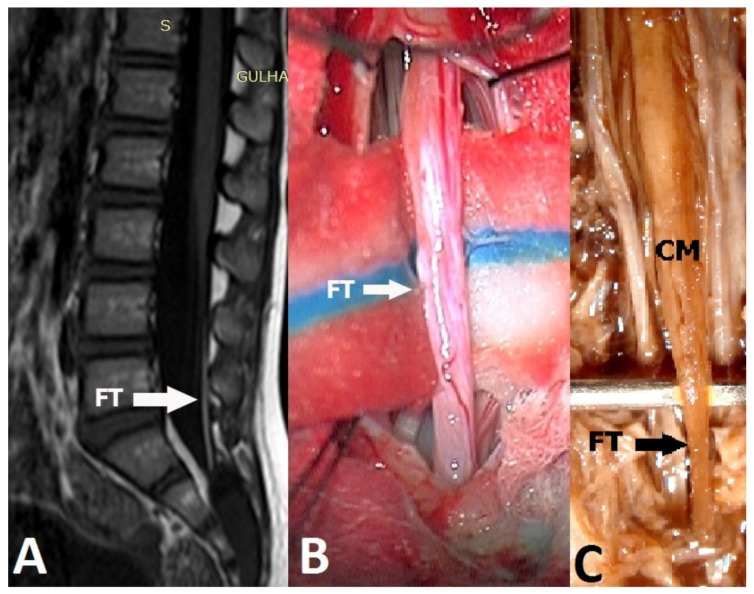
(**A**) T1-weighted sagittal MRI of a patient with fatty filum terminale (FT). (**B**) FT was cut and removed for histological examination. (**C**) Fetal FT was dissected and obtained from a fetus (CM: Conus medullaris).

**Figure 2 medicina-55-00133-f002:**
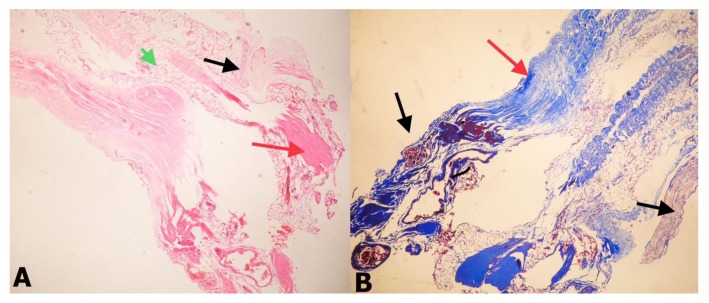
(**A**) Histopathological appearance of filum terminale obtained from a patient with tethered cord syndrome (TCS). Fibrosis (red arrow), adipose tissue (green arrow), and nerve fibers (black arrow) are obvious in the FT (H&E, ×40), (**B**) Masson trichrome staining. Black arrows indicate neural tissue and red arrow indicates collagen (Masson trichrome, ×40).

**Figure 3 medicina-55-00133-f003:**
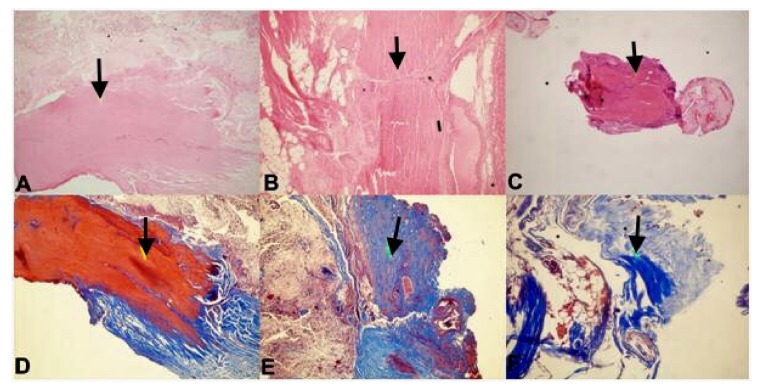
Fibrosis in fetal FT samples. Strong (**A**), moderate (**B**) and weak (**C**) fibrosis is observed in H&E staining (×10). Strong (**D**), moderate (**E**) and weak (**F**) fibrosis is also seen in Masson Trichrome staining (×10). Arrows indicate fibrosis areas.

**Figure 4 medicina-55-00133-f004:**
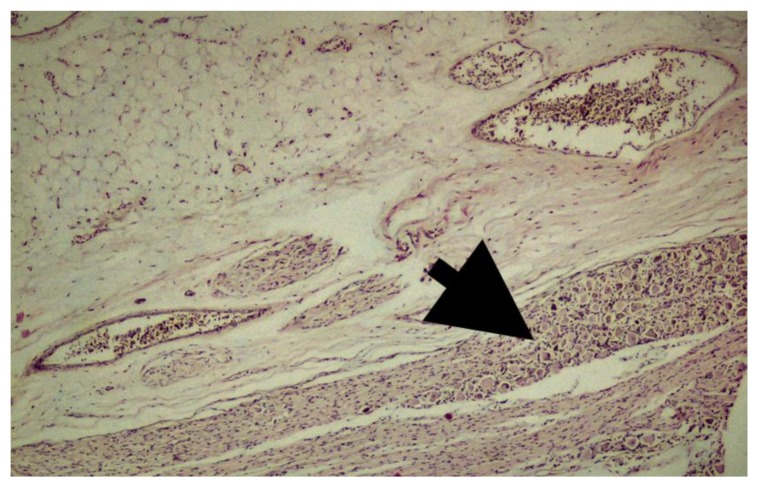
Fetal filum terminale composed of peripheral nerve and ganglion cells along with adipose and connective tissue (H&E, ×40). Thick black arrow indicates nerve ganglion bundle.

**Figure 5 medicina-55-00133-f005:**
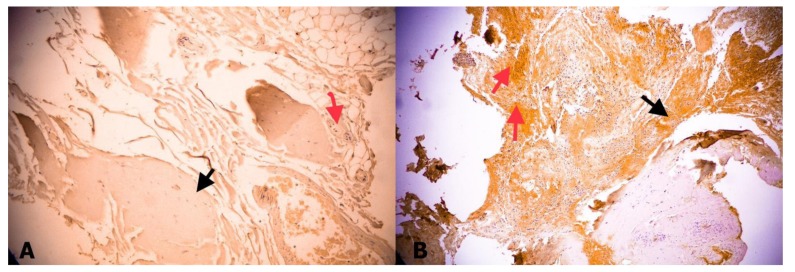
(**A**) No expression of sigma-GST in a fetal FT; black arrow indicates collagen, red arrow indicates neural tissue (GST, ×40). (**B**) Moderate expression of sigma-GST in FT of patient with TCS. Black arrow indicates collagen, red arrow indicates neural tissue (GST, ×40).

**Table 1 medicina-55-00133-t001:** Demographic and clinical features of the patients (Group 1). Sigma-class glutathione-S-transferase (GST) expression is shown in this table. The structure of the filum terminale is described based on radiological and/or surgical observation. Fibrosis is defined based on the histological examination.

Patient No	Age	Sex	Complaints	Associated Pathology	Level of Conus Medullaris	Structure of Filum Terminale	Fibrosis	GST Expression
**1**	12 years	F	Backache, walking disturbance	Fatty filum terminale	L5	Thick and fatty	+	+
**2**	3 years	F	Unable to walk	Type 1 SCM+Syringomyelia	L4	Normal	+	−
**3**	16 years	F	Backache	Syringomyelia	L4	Fatty	+	++
**4**	16 years	M	Back pain, left leg pain	None	L4	Normal	++	−
**5**	7 months	F	Hypertrichosis, scoliosis	Type 1 SCM	L4	Thick and fatty	+	+
**6**	6 years	F	Walking disturbance	Previously operated for myelomeningocele	L4	Normal	+++	++
**7**	2 years	F	Walking disturbance	Syringomyelia	L3	Normal	+	++
**8**	5 years	F	Urinary incontinence	None	L2	Normal	+	+
**9**	11 months	F	Walking disturbance	Type 1 SCM	L4	Normal	+	+
**10**	6 months	M	Motor weakness in left leg	Type 1 SCM	L4	Normal	++	+

F: Female; M: Male; SCM: Split cord malformation; (+ = weak; ++ = moderate; +++ = strong; − = negative).

**Table 2 medicina-55-00133-t002:** Fibrosis and sigma-class-GST expression in fetal FT samples (Group 2).

Fetus No	Fibrosis	GST-Expression
**1**	−	++
**2**	−	+
**3**	−	+
**4**	++	++
**5**	−	++
**6**	+++	−
**7**	+	−
**8**	+	−
**9**	++	+
**10**	+	−
**11**	+	−
**12**	−	+
**13**	−	+
**14**	−	+
**15**	+	−

+ = weak; ++ = moderate; +++ = strong; − = negative

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
