# Peer review of "Expression of Sigma-Class Glutathione-S-Transferase in Fetal and Pediatric Filum Terminale Samples: A Comparative Study"

_medicina, 2019, doi:10.3390/medicina55050133_

Round 1

Reviewer 1 Report

The manuscript by Kural et al., compared expression of Sigma-class glutathione-S-transferase (GST) in filum terminale (FT) from normal human fetuses and pediatric patients with tethered cord syndrome. The study used immunohistochemistry to compare GST expression between the two groups and histological staining to identify and score the level of fibrosis. No significant differences were observed between the two groups. The manuscript could be improved by including additional images and data and by careful revision and clarification of the results in the text.  This paper is recommended for publication following major revision.

Major points

·         In the introduction, the authors state that “If the viscoelasticity of FT is lost or disrupted by fibrosis, fatty infiltration or abnormal thickening, then caudal tension and traction may cause undue stress on the conus medullaris resulting in TCS” (Page 1-2, line 43-45). In Table 1, 7 out of 10 of the TCS patients have a normal filum terminale. The structure of the FT doesn’t seem to be a determining factor in TCS or have any correlation to the GST expression based on the data in Table 1? Could the authors comment on this?

·         In the methods section, the authors describe FT samples that were frozen and cut on a cryostat at 12-micron thickness for H&E and other stains, and FT samples that were paraffin-embedded, and cut at 4-micron thickness for H&E and immunohistochemistry. It is unclear whether frozen and paraffin blocks were obtained for both TCS FT samples and fetal FT samples. Please clarify this, as the way it is written suggests that the fetal FT samples were frozen, while the TCS FT samples were paraffin embedded. If this is the case it would make comparisons of immunohistochemical staining difficult, particularly when the sections were cut at different thicknesses. The authors should also list all the histological stains carried out in the Methods section.

·         In the results section, the authors state that ‘Fibrosis was observed in 8 FT from the fetuses (Table 2), while it was observed in all of ten FT samples from the children with TCS”. However, the structure of the FT in 7 TCS patients is listed as normal in Table 1. Is this description (thick and fatty, normal etc) based on macroscopic observations during surgery? Was fibrosis observed in FT samples that were otherwise described as having a ‘normal’ structure during microscopic analysis? Could the authors please comment on this?

·         The relative level of fibrosis is listed for the fetal FT samples (Table 2). This information should be provided for the TCS samples as well.

·         The authors state ‘’..fibrosis was moderate and strong in TCS samples. But it was weak in fetal FT samples” (Page 5-6, Line 128-134). However, according to Table 2 this was not always the case. Moderate or strong expression was noted in 3 samples (20%). Please clarify.

·         The authors state ‘Weak expression sigma-class GST was observed in 9 (60%) of fetal FT samples’. However, according to Table 2, weak GST expression (+) was only observed in 6 (40%) cases. Positive expression (including weak, moderate and strong) was reported in 9 (60%) of samples. The authors should clarify this point.

·         Micrographs should be improved. The background and contrast are inconsistent, this should be corrected particularly in Fig 4 when the authors are comparing immunostaining between different samples. Features mentioned in the figure legends or text such as fibrosis, nerve fibers etc (Figure 2) should also be labelled on the figures.

·         It would also be beneficial to show comparative images of the TCS and fetal samples with identical histological stains side by side. The results show a TCS sample stained with Masson Trichrome and a fetal FT sample stained for collagen. Additional images/figures should be provided to show H&E staining, Masson trichrome staining, and collagen staining of both fetal and TCS FT samples to make comparisons easier for the reader.

·         In the Discussion the authors state “Fibrosis and inelastic structure of FT, which is not present in fetal FT samples” (Page 7, Line 148). According to Table 2 fibrosis was present in 8 out of 15 (53%) fetal FT samples. This statement should be corrected or clarified.

·         The authors also state in the discussion (Page 7, Line 149-150) that “sigma-class GST expression was high in FT samples from the children with TCS.” The results in Table 1, however indicate that GST expression was negative (2 samples), weak (5 samples) and moderate (3 samples). There are no samples that were rated as having strong (+++) sigma-class GST expression in the FT samples obtained from the TCS patients. The authors even state in the results that “This expression [GST] is also weak in pediatric FT samples, and it was present in 8(80%) of 10 samples”. This should be revised.

Minor points

·         Table 1. The GST expression levels should be explained beneath the table ie. + = weak etc. This has been done for Table 2 but is missing in Table 1.

·         Text in the table also states “unable to walking”. This should be corrected to read “unable to walk.”

·         The written work could be improved, and spelling errors eg. Nomal (page 5, line 125) should be corrected.

Author Response

Major points

In the introduction, the authors state that “If the viscoelasticity of FT is lost or disrupted by fibrosis, fatty infiltration or abnormal thickening, then caudal tension and traction may cause undue stress on the conus medullaris resulting in TCS” (Page 1-2, line 43-45). In Table 1, 7 out of 10 of the TCS patients have a normal filum terminale. The structure of the FT doesn’t seem to be a determining factor in TCS or have any correlation to the GST expression based on the data in Table 1? Could the authors comment on this?

Our Reply: Thank you for your comment. In Table 1, the structure of filum terminale was reported as normal. But this is a surgical observation, not histological finding. During the surgery, we reported the gross appearance of filum terminale as normal, fatty or thick before cutting it. The filum terminales in normal appearance may have fibrosis or fatty tissue in histological examination. We agree with the reviewer that the structure of the FT based on surgical observation is not a determining factor for TCS. There is also many publications on this issue (Selcuki M, et al.Is a filum terminale with a normal appearance really normal?Childs Nerv Syst. 2003 Jan;19(1):3-10).

In the methods section, the authors describe FT samples that were frozen and cut on a cryostat at 12-micron thickness for H&E and other stains, and FT samples that were paraffin-embedded, and cut at 4-micron thickness for H&E and immunohistochemistry. It is unclear whether frozen and paraffin blocks were obtained for both TCS FT samples and fetal FT samples. Please clarify this, as the way it is written suggests that the fetal FT samples were frozen, while the TCS FT samples were paraffin embedded. If this is the case it would make comparisons of immunohistochemical staining difficult, particularly when the sections were cut at different thicknesses. The authors should also list all the histological stains carried out in the Methods section.

Our Reply: Sorry for our mistake. We did not use frozen technique. FT samples were only analyzed by formalin fixed paraffin embeddedblocks. Immunohistological staining was performed on slices obtained from the paraffin blocks. We corrected this issue in the Methods section.

In the results section, the authors state that ‘Fibrosis was observed in 8 FT from the fetuses (Table 2), while it was observed in all of ten FT samples from the children with TCS”. However, the structure of the FT in 7 TCS patients is listed as normal in Table 1. Is this description (thick and fatty, normal etc) based on macroscopic observations during surgery? Was fibrosis observed in FT samples that were otherwise described as having a ‘normal’ structure during microscopic analysis? Could the authors please comment on this?

Our Reply: Thank you for your comment. As we wrote in the first reply, in Table 1, the structure of FT is defined based on the preoperative radiological (Figure 1) and macroscopic surgical observation. As we wrote in the Results section that fibrosis was observed as a histological finding in all FT samples of pediatric TCS patients.

The relative level of fibrosis is listed for the fetal FT samples (Table 2). This information should be provided for the TCS samples as well.

Our Reply:We added a column for histological presence of fibrosis in FT samples from TCS in Table 1. 

The authors state ‘’..fibrosis was moderate and strong in TCS samples. But it was weak in fetal FT samples” (Page 5-6, Line 128-134). However, according to Table 2 this was not always the case. Moderate or strong expression was noted in 3 samples (20%). Please clarify.

Our Reply: We agree with the reviewer. We corrected this statement as “But it was mostly negative or weak in Group 2.”

The authors state ‘Weak expression sigma-class GST was observed in 9 (60%) of fetal FT samples’. However, according to Table 2, weak GST expression (+) was only observed in 6 (40%) cases. Positive expression (including weak, moderate and strong) was reported in 9 (60%) of samples. The authors should clarify this point.

Our Reply:We agree with the reviewer. We corrected this statement as “Weak expression sigma-class GST was observed in 6 (40%) of 15 Group 2 samples (Table 2).”

Micrographs should be improved. The background and contrast are inconsistent, this should be corrected particularly in Fig 4 when the authors are comparing immunostaining between different samples. Features mentioned in the figure legends or text such as fibrosis, nerve fibers etc (Figure 2) should also be labelled on the figures.

Our Reply: We corrected the micrographs. We also labeled the Figures.

It would also be beneficial to show comparative images of the TCS and fetal samples with identical histological stains side by side. The results show a TCS sample stained with Masson Trichrome and a fetal FT sample stained for collagen. Additional images/figures should be provided to show H&E staining, Masson trichrome staining, and collagen staining of both fetal and TCS FT samples to make comparisons easier for the reader.

Our Reply:Collagen was analyzed using H&E and Masson Trichrome staining. We did not perform immunostaining for collagen.

In the Discussion the authors state “Fibrosis and inelastic structure of FT, which is not present in fetal FT samples” (Page 7, Line 148). According to Table 2 fibrosis was present in 8 out of 15 (53%) fetal FT samples. This statement should be corrected or clarified.

Our Reply: We corrected this statement as “Fibrosis and inelastic structure of FT is obvious in 10 children with TCS”.

The authors also state in the discussion (Page 7, Line 149-150) that “sigma-class GST expression was high in FT samples from the children with TCS.” The results in Table 1, however indicate that GST expression was negative (2 samples), weak (5 samples) and moderate (3 samples). There are no samples that were rated as having strong (+++) sigma-class GST expression in the FT samples obtained from the TCS patients. The authors even state in the results that “This expression [GST] is also weak in pediatric FT samples, and it was present in 8(80%) of 10 samples”. This should be revised.

Our Reply: We revised this statement as “Sigma-class GST expression was frequent in FT samples from the children with TCS”. We also corrected other statement as “Weak expression sigma-class GST was observed in 6 (40%) of 15 Group 2 samples (Table 2). This expression is also mostly weak in Group 1, but frequent in 8(80%) of 10 samples.”

Minor points

Table 1. The GST expression levels should be explained beneath the table ie. + = weak etc. This has been done for Table 2 but is missing in Table 1.

Our Reply:We added this explanation in Table 1.

Text in the table also states “unable to walking”. This should be corrected to read “unable to walk.”

Our Reply: We corrected the errors.

he written work could be improved, and spelling errors eg. Nomal (page 5, line 125) should be corrected.

Our Reply:We corrected the errors.

Reviewer 2 Report

Dear Authors, 

The present study concluded preliminary results related to the expression of Sigma-class Glutathione-S-Transferase in Fetal and Pediatric Filum Terminale Samples. There are few concerns as shown below   

1. The present study lacking a clear objective in the introduction. 

2. There is a big concern for the consent from the family of patients whose samples were used in the study. It was not mentioned anywhere in the manuscript. If it has taken then it should be mentioned clearly in the method section of the manuscript.

3. Authors talk about many other related markers/pathways but did not provide any information related to those in their own experiments. Authors should provide the information of these markers in their present study by doing some more experiments (at least PGD2 data)  to get an exact idea and more clarity. 

4. In figures 2, 3, and 4, add any arrow or any other indication that can match with the information of figure legend. If possible, add high magnification images. 

Author Response

1.      The present study lacking a clear objective in the introduction.

Our Reply: In the last paragraph of Introduction section, we re-write the objective of our study.

2.      There is a big concern for the consent from the family of patients whose samples were used in the study. It was not mentioned anywhere in the manuscript. If it has taken then it should be mentioned clearly in the method section of the manuscript.

Our Reply:“We obtained informed consent from all patients’ families forthe surgical treatment of patient, as well as for the analysis of FT samples.” We wrote this statement at the beginning of the Methods section. 

3.      Authors talk about many other related markers/pathways but did not provide any information related to those in their own experiments. Authors should provide the information of these markers in their present study by doing some more experiments (at least PGD2 data)  to get an exact idea and more clarity.

Our Reply: As we wrote in the Limitations of our study that we could not investige PGD2 in FT samples due to lack of PGD2 kit. Once we provide this kit, we will study the expression of PGD2.

4.      In figures 2, 3, and 4, add any arrow or any other indication that can match with the information of figure legend. If possible, add high magnification images.

Our Reply:We inserted arrows to indicate the structures in the Figures. We also added some high magnification images.

Round 2

Reviewer 1 Report

Minor points

The authors should place the statement ‘Ethical approval for this study was obtained from the National Ethics Committee’ at the beginning of the Methods section. Most journals also require the ethics approval number to be provided.

In the methods section the details of the antibodies and reagents are listed except for GST-Sigma. This should be added. Page 5, line 116.

Figure 2. The figure legend states that ‘fibrosis, adipose tissue and nerve fibers are obvious in the FT’. These characteristics should be indicated on panel A. The arrows in panel B appear overly large and should be presented as is standard for micrograph labels.

Figure 3. Arrows difficult to see

Figure 5. No magnification is provided, and no scale bars are present on the micrographs. It doesn’t appear that the samples were imaged under the same conditions (constant exposure time, condenser opening, lamp brightness etc) which should be done if levels of staining are to be rated.

In the discussion the authors have now added the sentence ‘Fibrosis and inelastic structure of FT is obvious in 10 children with TCS.’  This should be amended to make it clear that this statement is referring to the microscopic analysis. Fibrosis was not always detected macroscopically (according to surgical observation).

Author Response

The authors should place the statement ‘Ethical approval for this study was obtained from the National Ethics Committee’ at the beginning of the Methods section. Most journals also require the ethics approval number to be provided.

Our reply: We wrote the approval number in the text, at the beginning of the methods section.

In the methods section the details of the antibodies and reagents are listed except for GST-Sigma. This should be added. Page 5, line 116.

Our reply: We added the details of the antibodies and reagants for GST-sigma in Methods section.

Figure 2. The figure legend states that ‘fibrosis, adipose tissue and nerve fibers are obvious in the FT’. These characteristics should be indicated on panel A. The arrows in panel B appear overly large and should be presented as is standard for micrograph labels.

Our reply: We corrected this figure. 

Figure 3. Arrows difficult to see

Our reply: We corrected this figure.

Figure 5. No magnification is provided, and no scale bars are present on the micrographs. It doesn’t appear that the samples were imaged under the same conditions (constant exposure time, condenser opening, lamp brightness etc) which should be done if levels of staining are to be rated.

Our reply: We corrected this figure and added magnification.

In the discussion the authors have now added the sentence ‘Fibrosis and inelastic structure of FT is obvious in 10 children with TCS.’  This should be amended to make it clear that this statement is referring to the microscopic analysis. Fibrosis was not always detected macroscopically (according to surgical observation).

Our reply: We corrected this sentence "Fibrosis and inelastic structure of FT is obvious in histological examination of 10 children with TCS." 

Reviewer 2 Report

Dear Authors,

I accept the manuscript in the current form for publication. 

Thanks

Author Response

Thank you